# Low *PRKAB2* Expression Is Associated with Poor Outcomes in Pediatric Adrenocortical Tumors, and Treatment with Rottlerin Increases the *PRKAB2* Level and Inhibits Tumorigenic Aspects in the NCI-H295R Adrenocortical Cancer Cell Line

**DOI:** 10.3390/cancers16061094

**Published:** 2024-03-08

**Authors:** Alcides Euzebio Tavares Xavier, Luciana Chain Veronez, Luís Fernando Peinado Nagano, Carolina Alves Pereira Correa, Mirela Baroni, Milena Silva Ramos, Rosane de Gomes de Paula Queiroz, Carlos Augusto Fernandes Molina, José Andres Yunes, Silvia Regina Brandalise, Sonir Antonio Rauber Antonini, Luiz Gonzaga Tone, Elvis Terci Valera, Carlos Alberto Scrideli

**Affiliations:** 1Departments of Genetics, Ribeirão Preto Medical School, University of São Paulo, 3000 Bandeirantes Avenue, Ribeirão Preto 14049-900, SP, Brazil; xavieralcides@usp.br (A.E.T.X.); luisnagano@usp.br (L.F.P.N.); carolinaalves.biologia@gmail.com (C.A.P.C.); mirellabaroni89@gmail.com (M.B.); milena.silva.ramos@usp.br (M.S.R.); lgtone@fmrp.usp.br (L.G.T.); 2Departments of Pediatrics, Ribeirão Preto Medical School, University of São Paulo, Ribeirão Preto 14049-900, SP, Brazil; lcveronez@usp.br (L.C.V.); rosane@fmrp.usp.br (R.d.G.d.P.Q.); antonini@fmrp.usp.br (S.A.R.A.); etvalera@hcrp.usp.br (E.T.V.); 3Department of Surgery and Anatomy, Ribeirão Preto Medical School, University of Sao Paulo, São Paulo, Ribeirão Preto 14049-900, SP, Brazil; molina@fmrp.usp.br; 4Boldrini Children’s Center, Campinas 13083-210, SP, Brazil; andres@boldrini.org.br (J.A.Y.); silvia@boldrini.org.br (S.R.B.); 5National Science and Technology Institute for Children’s Cancer Biology and Pediatric Oncology—INCT BioOncoPed, Porto Alegre 90035-003, RS, Brazil

**Keywords:** adrenocortical tumor, biomarker, AMPK, *PRKAB2*, Rottlerin

## Abstract

**Simple Summary:**

Adrenocortical tumors are rare neoplasms with an uncertain prognosis. A greater understanding of the biology of these tumors will allow new therapeutic targets to be identified and the prognosis of these patients to be better understood, enabling more precise therapeutic targeting. Our study sought to evaluate the expression of the *PRKAB2* gene as a prognostic biomarker in 63 cases of pediatric adrenocortical tumors and to analyze how efficient Rottlerin is in altering the tumorigenic profile of the NCI-H295R adrenocortical carcinoma (ACC) cell line.

**Abstract:**

Pediatric adrenocortical tumors (ACTs) are rare, highly heterogeneous neoplasms with limited therapeutic options, making the investigation of new targets with potential therapeutic or prognostic purposes urgent. The *PRKAB2* gene produces one of the subunits of the AMP-activated protein kinase (AMPK) complex and has been associated with cancer. However, little is known about the role AMPK plays in ACTs. We have evaluated how *PRKAB2* is associated with clinical and biological characteristics in 63 pediatric patients with ACTs and conducted in vitro studies on the human NCI-H295R ACC cell line. An analysis of our cohort and the public ACC pediatric dataset GSE76019 showed that lower *PRKAB2* expression was associated with relapse, death, metastasis, and lower event-free and overall survival rates. Multivariate analysis showed that *PRKAB2* expression was an independent prognostic factor when associated with age, tumor weight and volume, and metastasis. In vitro tests on NCI-H295R cells demonstrated that Rottlerin, a drug that can activate AMPK, modulated several pathways in NCI-H295R cells, including AMPK/mTOR, Wnt/β-catenin, SKP2, HH, MAPK, NFKB, and TNF. Treatment with Rottlerin decreased cell proliferation and migration, clonogenic capacity, and steroid production. Together, these results suggest that *PRKAB2* is a potential prognostic marker in pediatric ACTs, and that Rottlerin is promising for investigating drugs that can act against ACTs.

## 1. Introduction

Adrenocortical tumors (ACTs) have different incidences and behave distinctly in adult and pediatric patients. In adults, adrenocortical adenomas are relatively common and sometimes found accidentally, while adrenocortical carcinomas are rare, with an estimated incidence of 0.7 to 2 cases/year per million inhabitants [1]. In children, ACTs are very rare, with an estimated incidence of 0.2 to 0.3 cases/million inhabitants (aged < 20 years). Differentiating between malignant and benign forms is difficult, and, to date, no histological markers can reliably distinguish between them [2]. In the southern and southeastern regions of Brazil, the ACT incidence is around 4.2 cases per million children, which is 10–15 times higher than the world incidence of pediatric ACTs and has been associated with a specific germline mutation in the *TP53* gene (p.R337H), present in nearly 90% of cases [3,4,5].

The clinical presentation of patients with ACTs varies. Patients may be asymptomatic or present with symptoms related to mechanical effects stemming from tumor mass growth or excessive hormone production. Approximately 90% of pediatric cases have virilization, hypercorticolism, or both and are characterized as functioning [6,7,8]. Long-term survival is over 90% and 10% in patients with small localized tumors and metastatic disease, respectively. Around 50% of the patients with large localized tumors experience recurrence despite complete tumor resection [9], which reflects ACT heterogeneity. Complete tumor resection is the primary form of therapy, but it is not always possible [10]. In advanced ACT stages, therapy is based on using mitotane, an adrenolytic agent, combined with chemotherapeutics such as etoposide, doxorubicin, and cisplatin [9,11,12]. In an RNA sequencing analysis performed by our group, 14 ACT samples were sequenced, and 1215 differentially expressed genes were identified when the tumors of patients with good (*n* = 9) and poor clinical evolution (metastatic or recurrent cases or both, *n* = 5) were compared (GSE182022). To validate the expression profiles of hub genes, expression data from two public pediatric ACT datasets available in the Gene Expression Omnibus database (GSE76021 and GSE79019) were used. Interestingly, five hub genes were commonly validated between our cohort and the datasets. One of them was *PRKAB2*, which was downregulated in patients with a worse prognosis (metastatic and/or relapsed cases).

The *PRKAB2* gene produces the AMPKB2 protein, a regulatory unit within the AMPK (AMP-activated protein kinase) heterotrimeric complex. However, AMPKB2 also has a “scaffolding” function because it can mediate binding between the AMPK catalytic subunits and gamma regulator subunits to further stabilize the heterotrimeric complex [13].

Rottlerin, a natural compound extracted from plants, was initially described as a selective protein kinase C delta inhibitor [14]. Interestingly, Rottlerin also acts as a mitochondrial uncoupler (destabilizer), reducing cellular respiration, to increase the O_2_ level and to decrease the ATP level [15]. Rottlerin activates AMPK at the phosphorylation and activation (dissociation of catalytic subunits) levels [16]. The activation of AMPK and the PI3K/AKT/mTOR pathway induces autophagy and apoptosis in cancer cells [17], including the NCI-H295R cell line [18]. However, AMPK deregulation in pediatric ACTs has not been studied yet. Several dysregulated signaling pathways have been described in pediatric ACTs; for example, the dysregulation of the Wnt/β–catenin pathway [19] and the Hedgehog pathway [20], of genes such as *IGF2* and *IGF1R* [21], of *TP53* [22], and of the YAP1 oncogene [23] has been reported. The AMPK pathway negatively regulates *YAP*, a key gene in the Hippo signaling pathway [24]. Additionally, AMPK activation activates the tuberous sclerosis complex, which inhibits the activity of the mammalian target of rapamycin (mTOR) [25], whose signaling is activated in pediatric ACTs [26].

In this study, we have evaluated the clinical significance of the *PRKAB2* gene in pediatric ACTs and the functional effect of Rottlerin, an AMPK activator, in the NCI-H295R cell line. We will demonstrate that *PRKAB2* gene expression is associated with the clinical and biological characteristics of Brazilian children with ACTs and of the cohort available in the Children Oncology Group database. We have also evaluated how Rottlerin affects the NCI-H295R adrenal carcinoma cell line and will show that this compound stimulates *PRKAB2* and AMPK and thereby reduces essential functional activities and hormone biosynthesis in NCI-H295R cells.

## 2. Materials and Methods

### 2.1. Subjects

The present study analyzed *PRKAB2* gene expression in 63 pediatric adrenocortical samples after microdissection. The samples were obtained from pediatric patients treated at HCFMRP-USP (University Hospital of the Ribeirão Preto Medical School, University of São Paulo—Brazil) and Centro Infantil Boldrini (Campinas, state of São Paulo—Brazil). Samples of adrenal glands without neoplasms (*n* = 6) were obtained from children with Wilms tumors during nephrectomy, before any treatment. To exclude necrotic areas or adjacent normal tissue, fresh surgical tissues were immediately snap-frozen in liquid nitrogen, stored at −80 °C, and microdissected by an experimented pathologist before RNA was extracted. Only specimens that were at least 80% tumor cells were included in the study. In such samples, the possible invasion of the adrenal gland was excluded by conventional microscopy. This study was approved by the local Research Ethics Committee (CAEE: 65141422.2.0000.5440), and signed statements of informed consent were obtained from the children’s parents.

All the patients underwent clinical and hormonal evaluation by biochemical and imaging investigation. Abdominal and chest CT and bone scintigraphy were conducted at diagnosis and during follow-up to detect metastasis. The disease stage at diagnosis was based on the modified Sandrini classification of childhood ACTs. The patients comprised 50 girls and 13 boys; 45 patients were aged less than four years, and 18 patients were aged over four years. The tumor volume was less than and greater than 200 cm^3^ in 50 and 13 patients, respectively. The tumor weight was less than and greater than 100 g in 40 and 22 patients, respectively; there was no information about tumor weight for 1 patient. The germline *TP53* p.R337H mutation was evaluated by direct genomic DNA sequencing and was detected in 54 of the 63 patients (88.8%); 38, 14, 3, and 8 patients were classified as stages I, II, III, and IV, respectively, according to Sandrini’s modified staging [3]. Patient age > 4 years, tumor volume > 200 cm^3^, tumor weight > 100 g, tumor stage IV, and the presence of metastasis were associated with lower five-year event-free and five-year overall survival rates (Table 1 and Appendix A).

### 2.2. Cell Line

The NCI-H295R adrenocortical carcinoma cell line (ACTC^®^ CRL-2128™) was cultured in DMEM-F12 medium supplemented with 2.5% NuSerum (CORNING, Teo Oak Park, Bedford, MA, USA), 1% ITS (GIBCO, Grand Island, NY 14072, USA), 60 mg/mL penicillin, and 100 µg/mL streptomycin (GIBICO, Grand Island, NY 14072, USA) in a humid atmosphere containing 5% CO_2_ at 37 °C. An STR test was performed to confirm lineage authenticity. The H295R cell line presents an inactivated *TP53* gene due to the homozygous deletion of exons 8–9 and an activating mutation in the *CTNNB1* gene (c.T133C:p.S45P) and is the only commercially available confirmed human ACT.

### 2.3. Treatment with Rottlerin

NCI-H295R cells were treated with Rottlerin (protein kinase C delta inhibitor: 1610, Tocris Bioscience, Atlantic Road, Bristol, UK) at IC_50_ or IC_25_. A 1 mmol/L Rottlerin stock solution was prepared in dimethylsulfoxide (DMSO), and aliquots of this solution were stored at −20 °C. The IC_50_ and IC_25_ values were established by using Compusyn 1.0 software (https://www.combosyn.com/).

### 2.4. RNA Extraction and Gene Expression by Real-Time PCR (RT-qPCR)

Samples of ACTs and non-neoplastic adrenal cells were subjected to total RNA extraction by using the TRIZOL^®^ reagent (InvitrogenInc, Carlsdab, CA, USA); cDNA was synthesized with the High Capacity^®^ kit (AppliedBiosystems, Foster City, CA, USA) by following the protocol suggested by the manufacturer. After NCI-H295R cells were treated with Rottlerin for 48 h, they were subjected to RNA extraction and cDNA synthesis by using the same protocol mentioned above. Gene expression was analyzed by the qRT-PCR technique. The *GUSB* gene was used as an endogenous control, and the mean gene expression values of the six non-neoplastic adrenal samples were used as a calibrator. The employed primers are shown in Appendix A. Experiments were carried out on QuantStudio^TM^ 12K Flex equipment (Applied Biosystems, Foster City, CA, USA) by using the SYBR Green^®^ and GoTaq Probe^®^ PCR Master Mix (Applied Biosystems, Foster City, CA, USA): *GUSB* (4326320E), *STAR* (00986559), *MYCN* (00232074), *TCF7L2* (Hs01009044), *LEF1* (Hs01547250), *AXIN2* (00610344), *GSK3B* (Hs01047719), and *AKT1* (00178289). Relative gene expression was quantified by the 2^−ΔΔCT^ equation [27]. Standard curves were constructed for each gene to determine the efficiency. All the curves showed slopes between −3.6 and −3.3 (efficiency of 90–100%). All the samples were analyzed in at least three independent experiments, performed in triplicate.

### 2.5. Cell Viability Assay

Cell viability was analyzed by using the CellTiter Glo^®^ Cell Luminescent Cell Viability kit (Promega, Madison, WI, USA); the manufacturer’s protocol was followed. NCI-H295R cells were cultivated in a 96-well plate at a density of 2 × 10^4^ cells/well; luminescence was read after treatment with Rottlerin at 4.5 µM (IC_50_) or 2.25 µM (IC_25_) for 24, 48, or 72 h. At least three independent experiments were performed for each concentration and time in triplicate.

### 2.6. Clonogenic Assay

NCI-H295R cells were treated with Rottlerin at 4.5 µM (IC_50_) or 2.25 µM (IC_25_) in 25 cm^3^ bottles at a confluence of 4 × 10^6^ for 24 or 48 h. Then, 2.5 × 10^2^ cells/well were seeded in a 12-well plate and cultured for four weeks. The medium was changed every 48 h. Colonies were fixed with methanol and stained with Giemsa. Colonies with over 50 cells were considered for analysis. The assays were analyzed in at least three independent experiments, performed in triplicate.

### 2.7. Transwell Assay

NCI-H295R cells were treated with Rottlerin at 4.5 µM (IC_50_) or 2.25 µM (IC_25_) in 25 cm^3^ bottles at a confluence of 4 × 10^6^ for 48 h. Then, 2 × 10^5^ cells suspended in DMEM-F12 medium without supplementation were seeded per insert. Supplemented medium was added to the 24-well plate, below the insert, to stimulate cell migration. Cells were allowed to migrate for 72 h. Next, they were fixed with methanol and stained with Giemsa. The area of each scratch was photographed under an inverted microscope in three randomly selected fields along the scraped line and measured by using ImageJ software version 1.48 (NIH, Washington, DC, USA). The assays were repeated in three independent experiments in triplicate.

### 2.8. Hormone Measurement

NCI-H295R cells, 5 × 10^5^ cells/well, were seeded in a six-well plate for 24 h. The cells were treated with DMSO or Rottlerin at 4.5 µM (IC_50_) or 2.25 µM (IC_25_) for 48 h. After that, the culture medium was collected and immediately stored at −80 °C. Cortisol, testosterone, and D4-androstenedione concentrations were measured in the supernatant of the NCI-H295R cell medium by a radioimmunoassay (RIA), as previously described [28]. All the hormones were measured in triplicate, and the mean was normalized to the cell viability effects in the same treatment conditions.

### 2.9. Western Blotting

After treatment with Rottlerin at 4.5 µM (IC_50_) or 2.25 µM (IC_25_) for 6, 12, 24, or 48 h, between 20 and 40 μg of proteins was separated by SDS-PAGE electrophoresis and incubated with 1:1000 anti-AMPKβ2 primary antibodies (4148S), 1:1000 p-AMPK (#2535), 1:1000 AMPKα-total (#2532), 1:1000 p-mTOR (#2971), and 1:1000 mTOR-total (#2983), obtained from Cell Signaling Technology (Danvers, MA, USA), and 1:2500 anti-GAPDH (sc-47724) and 1:500 MAP LC3B (sc-271625), obtained from Santa Cruz Biotechnology (Santa Cruz, CA, USA). Anti-mouse IgG 1:5000 (#7076) and anti-rabbit IgG 1:5000 (#7074) secondary antibodies were obtained from Cell Signaling Technology (Danvers, MA, USA). A molecular weight marker (MW) and the protein of interest were run on the same gel. The MW was detected by colorimetry to control the size of the band of interest. The protein was detected by chemiluminescence; the ECL^TM^ Western Blotting Analysis System (Amersham GE Healthcare, Buckinghamshire, UK) was used. Intensity was quantified by using ImageJ software version 1.48 (NIH, Washington, DC, USA), relativized by the control, and normalized relative to an endogenous protein (GAPDH). Membrane stripping was performed a maximum of three times by using Restore Western Blot Stripping buffer (Thermo Scientific, Rockford, IL, USA).

### 2.10. RNA Sequencing (RNAseq)

To investigate the role played by Rottlerin in NCI-H295R cells, global gene expression by RNAseq analysis was measured in cells treated with Rottlerin (at IC_50_) or the control (DMSO). The Illumina Stranded mRNA Sample Preparation kit (Illumina Inc., San Diego, CA, USA) was used to prepare the samples. The generated libraries were evaluated by using Tapestation equipment (Agilent Technologies, Santa Clara, CA, USA) and quantified on a Qubit fluorimeter with the Qubit dsDNA BR Assay kit (Life Technologies, Carlsbad, CA, USA). Clustering and sequencing were performed on the Illumina NovaSeq 6000 equipment by using the NovaSeq SP Kit (200 cycles) (Illumina Inc.), generating 800 million reads per sequencing run. The data quality of the two samples (Rottlerin at IC_50_ and DMSO) was evaluated by using the FastQC (0.11.3) and MultiQC (v1.10) programs, which provide information about basic sequence statistics, quality, GC content, sequence size distribution, sequence duplication levels, and over-represented sequences. To align, to map, and to count the sequences against the reference genome, the STAR aligner (v2.7.8a) was used. From the count of reads in each sample, the R programming language (v4.1.2) was applied; more specifically, the DESeq2 package was employed, which enabled differential expression between the different conditions of the experiment to be analyzed. On the basis of LogFC, the differentially expressed genes (DEGs) were separated into up- and downregulated genes and subsequently enriched for deregulated pathways through the enrichment platform. A LogFC of 1.5 and a *p*-value < 0.05 were considered. Sequencing was carried out at the Institute for Cancer Research (IPEC), Guarapuava—PR. These data were deposited in the Gene Expression Omnibus database (GSE255729)

### 2.11. Statistical Analysis

Functional assays and patients’ samples were statistically analyzed by using the GraphPadPrism 8.0 (GraphPad Software, San Diego, CA, USA) and SPSS 20.0 (SPSS Inc. Chicago, IL, USA) statistical programs. At least three independent experiments were carried out in triplicate; the average of the experiments was considered. The Mann–Whitney test was used to analyze patients’ gene expression and clinical characteristics. One-way and Two-way ANOVAs with a post hoc Bonferroni test were used for functional tests; *p* < 0.05 was adopted as the level of statistical significance. The analysis of five-year event-free survival (with relapse or death from any cause being considered an unfavorable event) and five-year overall survival (with death being considered an unfavorable event) rates were based on Kaplan–Meier curves and log-rank tests; the median *PRKAB2* gene expression was used as the cut-off point for high or low expression. The multivariate test using the Cox regression model was applied to assess the independence of the factors in the predictive capacity.

## 3. Results

### 3.1. The PRKAB2 Gene Expression Profile in the Brazilian Cohort

In our cohort, lower *PRKAB2* gene expression was associated with shorter overall (Figure 1A, *p* = 0.004) and event-free survival (Figure 1B, *p* < 0.001). Lower *PRKAB2* gene expression was also associated with death (Figure 1C, *p* = 0.0066), tumor relapse (Figure 1D, *p* = 0.0018), and Sandrini IV staging (Figure 1E, *p* = 0.0197) cases.

To further validate our findings, we analyzed the *PRKAB2* gene expression in an independent Children Oncology Group cohort deposited in the Gene Expression Omnibus database (GSE76019) (*n* = 34). In this cohort, germline *TP53* mutations were detected in 23 (67.6%) patients (10 *R337H*, 13 other *TP53* mutations, 10 WT, and 1 patient without information). As in the case of our cohort, low *PRKAB2* gene expression was associated with shorter overall (Figure 2A, *p* = 0.044) and event-free survival (Figure 2, *p* = 0.040); the median of the expression was used as the cut-off point for high or low expression. In addition, *PRKAB2* underexpression was associated with poor clinical outcomes: this gene was less expressed in cases of death (Figure 2, *p* = 0.0116), unfavorable clinical events (recurrence, death, or metastasis) (Figure 2, *p* = 0.0094), and Sandrini IV staging (Figure 2, *p* = 0.0219).

Multivariate analysis showed that lower *PRKAB2* gene expression was a significant independent prognostic factor for both overall survival (*p* = 0.006) and event-free survival (*p* = 0.003) when analyzed in combination with classic prognostic factors such as the patient’s age (greater or less than four years), tumor volume (greater or less than 200 cm^3^), tumor weight (greater or less than 100 g), and presence and absence of metastasis (Figure 3A,B).

### 3.2. Cell Viability

Rottlerin decreased NCI-H295R cell proliferation in a dose- and time-dependent manner (Figure 4A), with IC50 and IC25 calculated as 4.5 µM and 2.25 µM, respectively. Figure 4B shows cell viability after treatment with Rottlerin at IC_50_ (4.5 µM, *p* = 0.0034) or IC_25_ (2.25 µM) for 48 h compared to the control (DMSO).

### 3.3. Transwell Assay

The Transwell assay showed that NCI-H295R cells had decreased migration capacity after they were treated with Rottlerin at IC_50_ or IC_25_ (*p* = 0.025) compared to the control (DMSO) (Figure 4C,D); the migration time was 72 h.

### 3.4. Clonogenic Assay

Treatment with Rottlerin at IC_50_ or IC_25_ for 24 h (*p* = 0.0102) or 48 h (*p* = 0.0102) reduced the clonogenic capacity of NCI-H295R cells compared to the control (DMSO) (Figure 4E–G).

### 3.5. Rottlerin Upregulates PRKAB2 in NCI–H295R Cells

NCI-H295R cells treated with Rottlerin showed increased *PRKAB2* gene and PRKAB2 protein expression. Treatment was carried out for different times: 6, 12, 24, or 48 h. The highest AMPKB2 protein expression level occurred after treatment with Rottlerin at IC_50_ for 6 h, when the protein level doubled compared to the control (DMSO). After treatment under these conditions for 48 h, the AMPKB2 protein level increased by 70% compared to the control (DMSO) (Figure 5A). Moreover, *PRKAB2* gene expression increased after treatment with Rottlerin at IC_50_ (*p* = 0.0063) or IC_25_ (*p* = 0.0287) for 48 h (Figure 5B).

### 3.6. Rottlerin Modulates via AMPK/mTOR, Suppresses the Steroidogenic Factor, Reduces Hormone Biosynthesis, and Stimulates Autophagy in NCI-H295R Cells

Rottlerin demonstrated the potential for activating total AMPK (alpha subunits) and phosphorylating AMPK at Thr-172 in NCI-H295R cells. Treatment with Rottlerin at IC_50_ for 6 h activated AMPK more effectively, while treatment for 12 h decreased the concentration of this protein, treatment for 24 h promoted stabilization, and treatment for 48 h further decreased the AMPK level (Figure 6A).

The expression of *STK11*, the gene underlying the production of liver kinase LKB1, the main pathway of AMPK phosphorylation at Thr-172, increased after treatment with Rottlerin at IC_25_ for 48 h (Figure 6C, *p* = 0.0033). In addition, treatment with Rottlerin at IC_50_ for 48 h reduced the level of acetyl-CoA carboxylase (ACC), a marker of AMPK activity. However, AMPK expression was lower at all times compared to the control (DMSO) (Figure 6B). Also, the gene expression of the steroidogenic factor STAR was reduced after treatment with Rottlerin at IC_50_ (*p* = 0.0280) or IC_25_ (*p* = 0.0061) (Figure 6E). Cortisol, D4-androstenedione, and testosterone secretion by the treated NCI-H295R cells was also lower (Figure 7).

Treatment with Rottlerin at IC_25_ or IC_50_ for 12, 24, or 48 h reduced the total mammalian target of rapamycin (mTOR) and p-mTOR activity. However, treatment for 6 h stimulated total mTOR and p-mTOR (Figure 6B). Furthermore, treatment with Rottlerin at IC_50_ for 48 h stimulated the expression of the *TSC1* gene, described in the mTOR inhibition cascade (*p* = 0.0057) (Figure 6D).

Compared to the control (DMSO), treatment with Rottlerin for 24 or 48 h increased the level of the MAP LC3B protein, an autophagy marker (Figure 6F). The LC3B:LC3-I cleavage is an intermediate cleavage that occurs prior to binding to the autophagosome; LC3B:LC3-II is the second cleavage and the final version, which can bind to the autophagosome. Both cleavages were upregulated after treatment with Rottlerin (Figure 6F,G), but LC3B:LC3-II expression was more significantly different. Additionally, the BAX protein level increased compared to the control: higher levels of this protein are correlated with increased apoptosis.

### 3.7. Rottlerin Inhibits the Wnt/β-Catenin Pathway

Treatment with Rottlerin decreased the β-catenin level in NCI-H295R cells. Cells treated with Rottlerin at IC_50_ or IC_20_ for 12, 24, or 48 h showed lower β-catenin and active β-catenin levels compared to the control (DMSO) (Figure 8). In addition, treatment with Rottlerin at IC_25_ decreased the expression of key genes in the Wnt/β-catenin pathway, such as *MYCN* (*p* = 0.0012), *TCF7* (*p* ≤ 0.0001), *LEF1* (*p* = 0.254), and *AXIN2* (*p* ≤ 0.0001), and treatment with Rottlerin at IC_50_ decreased *WNT3* gene expression. Treatment with Rottlerin at IC_50_ for 48 h increased *GSK3B* gene expression (Figure 8).

### 3.8. Treatment with Rottlerin for 48 h Potently Inhibits SKP2 Protein Expression

Rottlerin potently inhibited SKP2 protein expression. The highest inhibition rate was achieved after treatment with Rottlerin at IC_50_ for 48 h. However, inhibition was 50% after treatment with Rottlerin at IC_25_ for 12 or 24 h and 90% after treatment with Rottlerin at IC_25_ for 48 h (Figure 9).

### 3.9. Transcriptome Analysis after Treatment with Rottlerin

#### Rottlerin Stimulates Key Genes That Suppress the Hedgehog Pathway

RNAseq of NCI-H295R cells treated with Rottlerin at IC_50_ for 48 h helped us understand how Rottlerin affected the cells. After normalizing and enriching the comparative DMSO × IC_50_ data, we found that several cellular pathways and gene ontology were altered (Figure 10).

Among the positively regulated pathways, we can mention the Hedgehog development pathway (hh), which had 10 differentially expressed genes, including *GRK3*, *EVC*, *SMO*, *KIF3A*, *SUFU*, *GPR161*, *BOC*, *ARRB1*, *BTRC*, and *CDON* (*p* = 0.003) (Figure 11A).

On the other hand, some pathways, such as the MAPK pathway, were negatively regulated: 30 genes were negatively regulated, namely, *FLT1*, *TGFA*, *HSPB1*, *AREG*, *TNF*, *DUSP10*, *FGF9*, *MYC*, *CD14*, *FGF2*, *DUSP4*, *DUSP5*, *HSPA8*, *JUN*, *DUSP2*, *JUN*, *GADD45B*, *GADD45*, *DUSP1*, *HSPA6*, *VEGFC*, *FOS*, *DUSP8*, *GADD45G*, *NR4A1*, *DDIT3*, *FGF18*, *HSPA1B*, *ATF4*, and *HSPA1A* (*p* = 0.00000048) (Figure 11B). The NFKB pathway was also negatively regulated: the 13 negatively regulated genes were *CXCL8*, *GADD45B*, *GADD45A*, *DDX5*, *CXCL*, *TNF*, *CXCL2*, *GADD45G*, *NFKBIA*, *CSNK2B*, *LTA*, *CD14*, and *BIRC3* (*p* = 0.002) (Figure 11C). The TNF pathway had 16 negatively regulated genes: *JUN*, *VEGFC*, *PIK3R2*, *CXCL1*, *CXCL2*, *NFKBIA*, *FOS*, *TNF*, *CREV1*, *BCL3*, *LTA*, *CCL2*, *JUNB*, *ATF4*, *BIRC3*, and *CREB5* (*p* = 0.0000033) (Figure 11D).

## 4. Discussion

Pediatric ACTs have an embryonic origin and limited therapeutic options, especially for more advanced stages of the disease [29]. ACTs are highly recurrent, even after resection, which is the only effective treatment proposed to date. The overall five-year survival rate is around 54% to 70%, and this rate is less than 20% for metastatic disease [30]. Therefore, investigating new targets with potential therapeutic or prognostic purposes is urgent because treatments for pediatric ACTs are currently limited, and the prognosis is uncertain. In this study, we observed that the *PRKAB2* gene is a potential prognostic marker in ACT, and that cases of death, relapse, and metastasis at diagnosis within our cohort and the Children Oncology Group (COG) cohort (GSE76019) had a lower expression of this gene. In addition, in both cohorts, lower *PRKAB2* gene expression was associated with lower five-year overall and five-year event-free survival rates.

Predicting the prognosis and the risk of relapse in children with ACTs remains challenging because the anatomopathological parameters on which prediction relies are inaccurate. Currently, the most commonly used prognostic classifications in pediatric patients classify these tumors into stages I (less aggressive) to IV (more aggressive) and consider the degree of tumor resection, tumor weight, invasion of contiguous tissues or metastasis, and hormone levels after tumor resection. However, none of the available classifications can completely determine the prognosis, especially in intermediate stages [3,4]. Patients at the extremes of staging have a more definite prognosis, which is very favorable in stage I and very unfavorable in stage IV [3,4]. Low *PRKAB2* gene expression was correlated with Sandrini IV staging (*n* = 8) in both our cohort and the COG cohort (GSE76019). Originally, the Brazilian cohort was classified according to the modified Sandrini staging; however, to evaluate other staging systems already proposed on the basis of *PRKAB2* gene expression, we classified this cohort into the ENSAT, IPACTR, and IPACTR-II stages. The survival rates were similar, except for ENSAT, without patients in stage III (Appendix A). Furthermore, in our cohort, lower *PRKAB2* gene expression was correlated with ENSAT (*n* = 9), IPACTR (*n* = 7), and IPACTR-II (*n* = 8) IV staging (Appendix A).

In addition to this, when we evaluated *PRKAB2* gene expression by multivariate analysis with classic prognostic factors such as patient age and tumor size and volume, we found that this gene was an independent factor in the prognostic prediction of five-year overall and five-year event-free survival rates in pediatric patients with ACTs.

The therapeutic options to treat ACT, especially its more advanced stages, are scarce and barely improve patient survival. In this scenario, Rottlerin has emerged as a potential agent against cancer: Rottlerin acts as a mitochondrial uncoupler, decreasing the intracellular ATP level and activating AMPK by inhibiting mTOR, thereby regulating autophagic and cell growth pathways [25,31]. AMPK activation stimulates genes that produce its α, β, and γ subunits. *PRKAB2*, the object of this study, is among these genes and underlies the production of the AMPKB2 subunit, which constitutes the AMPK heterotrimeric complex. Thus, the reduced level of a subunit can directly interfere with AMPK activity. Treatment with Rottlerin at 4.5 µM (IC_50_) or 2.25 µM (IC_25_) for 24 or 48 h reduced the clonogenic capacity, migration, and viability of NCI-H295R cells.

Treatment with Rottlerin increased total and phosphorylated AMPK activity, which peaked after treatment with Rottlerin for 6 h. AMPK has a heterotrimeric conformation and functions as a cellular energy sensor. ATP depletion and an AMP increase change the AMPK conformation, exposing its phosphorylation loop and allowing the dissociation of the α1 or α2 catalytic unit [13,25,32]. LKB1 kinase is the main agent of AMPK phosphorylation at Thr-172. The expression of the *STK11* gene, coding for LKB1, was significantly upregulated after treatment with Rottlerin for 48 h.

Rottlerin also reduced total and phosphorylated mTOR protein expression in NCI-H295R cells. The effect was greater after treatment for 48 h. AMPK induces TSC1, inhibiting mTOR [33,34]. Interestingly, treatment with Rottlerin for 48 h induced the *TSC1* gene, suggesting that AMPK activation in NCI-H295R cells by Rottlerin stimulates TSC1 via mTOR inhibition, and mTOR signaling has been shown to be activated in pediatric ACTs [26]. In line with AMPK activation and mTOR inhibition, treatment with Rottlerin increased the LC3-I and II protein levels, suggesting that the AMPK/mTOR pathway may induce autophagy after the treatment of NCI-H295R cells with Rottlerin. The BAX protein level was also higher, suggesting an effect on apoptosis activation.

As the AMPK activity increased, ACC decreased regardless of the treatment time. ACC is a precursor of lipid metabolism and an important factor in hormone production. AMPK inhibits ACC, so ACC is used as a marker of AMPK activity [35,36,37]. Interestingly, the *STAR* steroidogenic factor mRNA level, a marker of steroid production, was significantly reduced. Excessive hormone production is a common feature of pediatric ACTs. When the tumor is functioning (producing excess hormone), it causes virilization, Cushing syndrome, and other secondary effects in pediatric patients [38,39]. Accordingly, NCI-H295R cells treated with Rottlerin had significantly decreased cortisol, D4-androstenedione, and testosterone levels, also suggesting that this drug can be used to suppress the effects of excess steroids in ACT patients.

The Wnt/β-catenin pathway is activated in pediatric ACTs, which is evidenced by β-catenin accumulation in the cell nucleus [19]. GSK3B underlies β-catenin phosphorylation, and its activity is related to the initiation of β-catenin degradation [40]. Our results showed that treatment with Rottlerin at IC_50_ or IC_25_ for 6 h increased the GSK3B and p-GSK3B protein levels, but these levels declined thereafter. Interestingly, treatment with Rottlerin at IC_50_ elevated the mRNA levels of the *GSK3B* gene, suggesting that post-transcriptional regulation occurred. Additionally, treatment with Rottlerin at IC_25_ or IC_50_ for 48 h reduced the expression of the *AKT* gene, which underlies the inhibition of phosphorylated and non-phosphorylated GSK3B. In mTOR/AKT signaling, TSC1/TSC2 is inhibited by AKT and ultimately inhibits mTORC1 [41]. Thus, our results suggest that Rottlerin acts in several cell signaling pathways in in vitro assays with NCI-H295R cells, independently or by crosstalk.

The *SKP2* gene has already been described in several tumorigenic processes [42,43]. Its high expression has already been correlated with colorectal cancer, breast cancer, and cervical cancer, with such high expression being associated with the degree of malignancy [44]. Also, a study on skeletal muscle atrophy has demonstrated that AMPK inhibition in skeletal muscle cells decreases autophagy and increases SKP2 protein expression [45]. In our study, treatment with Rottlerin at either of the tested concentrations reduced SKP2 protein expression after 12 h, and this reduction became more evident after treatment for 48 h. In addition, such treatment stimulated autophagy. Thus, we can suggest that AMPK activity inhibits SKP2 in ACT cells. However, further studies are needed to understand the role played by the AMPK/SKP2 axis in ACTs.

Other pathways have been reported to interact with AMPK in ACTs or in the tumorigenic process of other types of cancer, including the Hedgehog signaling pathway [20], the MAPK signaling pathway [45], tumor necrosis factor (TNF) [46], and nuclear factor kappa B (NFKB) [47,48].

Interestingly, these pathways were also enriched during transcriptome analysis in the NCI-H295R cell line after treatment with Rottlerin. We observed that these pathways were modulated after treatment with Rottlerin at IC_50_ compared to the control (DMSO). Among the upregulated developmental pathways, the Hedgehog pathway is the most significant. Among the Hedgehog pathway ligands, SHH is deregulated in ACTs in both adults and children [20]. AMPK’s association with the Hedgehog pathway has opposite effects on different tumors. In medulloblastoma cell lines, AMPK activation inhibits the transcriptional activity induced by the canonical SHH pathway through GLI1 phosphorylation [35]. In contrast, the activation of *SMO*, a gene that was upregulated in our RNASeq analysis, activates autophagy via LBK1-AMPK in the non-canonical Hedgehog pathway [49]. It is noteworthy that some of the genes associated with the Hedgehog pathway and upregulated in our RNASeq analyses have been described as important for neoplastic cell survival. *CDON* gene inhibition appears to promote a selective advantage in SHH-expressing tumors by blocking CDON-induced apoptosis [45]. Different GSK3 modulators have been suggested to trigger distinct targets that can induce or inhibit tumor cell proliferation depending on the tumor type [50]. New studies investigating this complex relationship between AMPK and the Hedgehog pathway are needed.

We found that the MAPK, NFKB, and TNF pathways were downregulated in NCI-H295R cells after treatment with Rottlerin. MAPK signaling is upregulated in several types of tumors, including malignant ACTs, and inhibiting this pathway decreases proliferation and steroid production in NCI-H297R cells [51]. A profound and complicated interaction between AMPK and MAPK has been described, demonstrating that there is an inhibitory crosstalk between MAPK and AMPK [37]. In addition to the MAPK pathway, the NFKB and TNF pathways were also downregulated in NCI-H295R cells after they were treated with Rottlerin. Studies have already described that AMPK inhibits NFKB and TNF [52,53,54,55]. However, our transcriptome results must be analyzed with caution, as they were performed in single replicates.

A potential bias of this study was the use of a single ACT cell line. However, NCI-H295R is the only commercially available strain. Another available adrenal tumor cell line is SW13, but it does not produce steroids, and it is not known whether the cell line is derived from a primary ACT or, more likely, from a metastatic tumor that has spread to the adrenal cortex [56]. For this reason, its use in in vitro studies has been discouraged.

## 5. Conclusions

Our results have shown that lower *PRKAB2* gene expression is a potential marker of a worse prognosis in pediatric ACTs with an independent prediction capacity. Furthermore, the treatment of the NCI-H295R cell line with Rottlerin, a promising agent against ACTs, stimulates *PRKAB2* and total and phosphorylated AMPK; inhibits the mTOR, Wnt/β-catenin, and SKP2 pathways; stimulates autophagy and apoptosis; and decreases cell viability, clonogenic capacity, and migration in in vitro assays. Treatment with Rottlerin also alters the mRNA levels of important pathways for the tumorigenic process, such as the Hedgehog, MAPK, NFKB, and TNF pathways. These results are promising, but further studies are needed to understand *PRKAB2* in ACT cells and to determine on which molecular mechanisms Rottlerin acts.

## Figures and Tables

**Figure 1 cancers-16-01094-f001:**
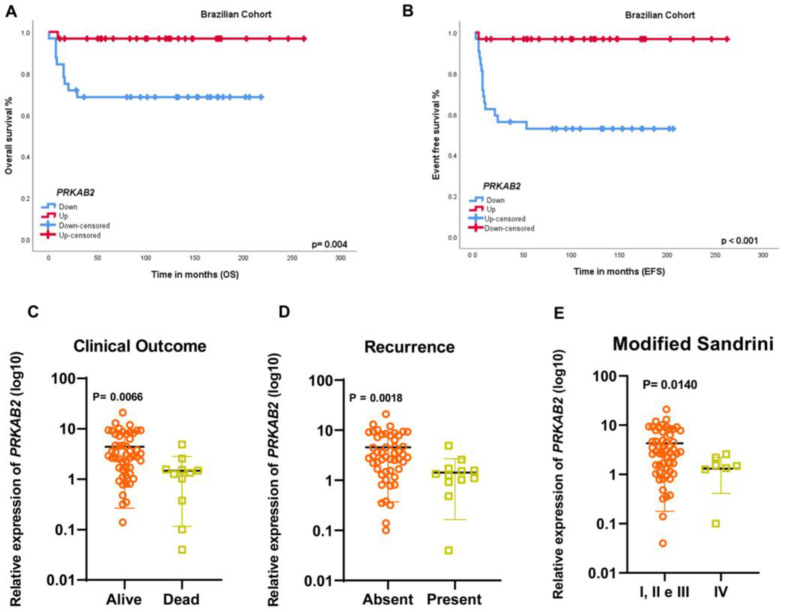
(**A**) Kaplan–Meier curves and log-rank tests showing that patients with shorter overall survival (68.6%) had lower *PRKAB2* gene expression compared to patients with longer overall survival (96.8%) (*p* = 0.004). (**B**) Patients with shorter event-free survival (52.9%) had lower *PRKAB2* gene expression compared to patients with longer event-free survival (96.8%) (*p* < 0.001). (**C**) Patients who died had lower *PRKAB2* gene expression (*p* = 0.0056) (Mann–Whitney test). (**D**) Relapsed patients had lower *PRKAB2* gene expression (*p* = 0.0018) (Mann–Whitney test). (**E**) Patients with Sandrini IV staging showed lower *PRKAB2* gene expression (*p* = 0.0197) (Mann–Whitney test).

**Figure 2 cancers-16-01094-f002:**
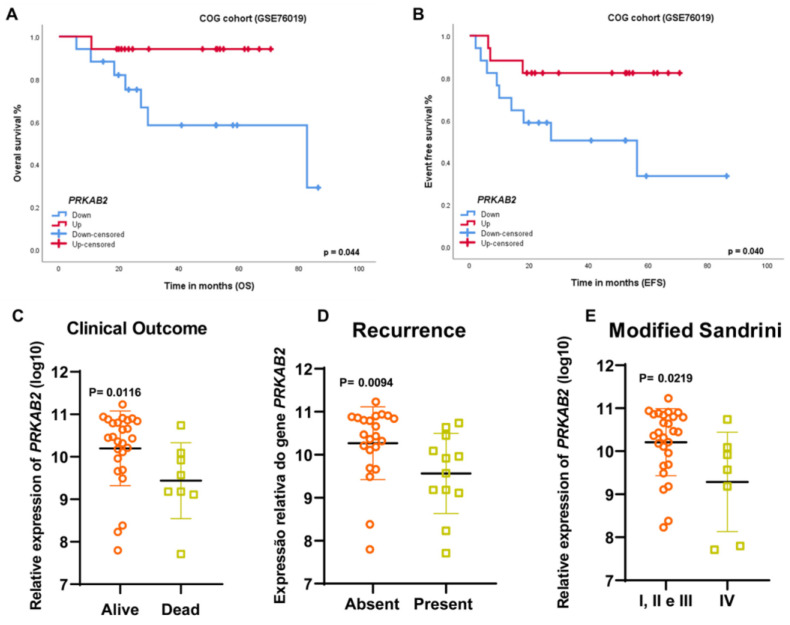
(**A**) Kaplan–Meier curves and log-rank tests in the COG cohort showing that patients with shorter overall survival (*n* = 17) had lower *PRKAB2* gene expression compared to patients with longer overall survival (*n* = 17) (*p* = 0.044). (**B**) Patients with shorter event-free survival (*n* = 17) had lower *PRKAB2* gene expression compared to patients with longer event-free survival (*n* = 17) (*p* = 0.040). (**C**) Patients who died showed lower *PRKAB2* gene expression (*p* = 0.0116) (Mann–Whitney test). (**D**) Patients who presented recurrence followed by metastasis showed lower *PRKAB2* expression (*p* = 0.0094) (Mann–Whitney test). (**E**) Patients with Sandrini IV staging showed lower *PRKAB2* gene expression (*p* = 0.0219) (Mann–Whitney test).

**Figure 3 cancers-16-01094-f003:**
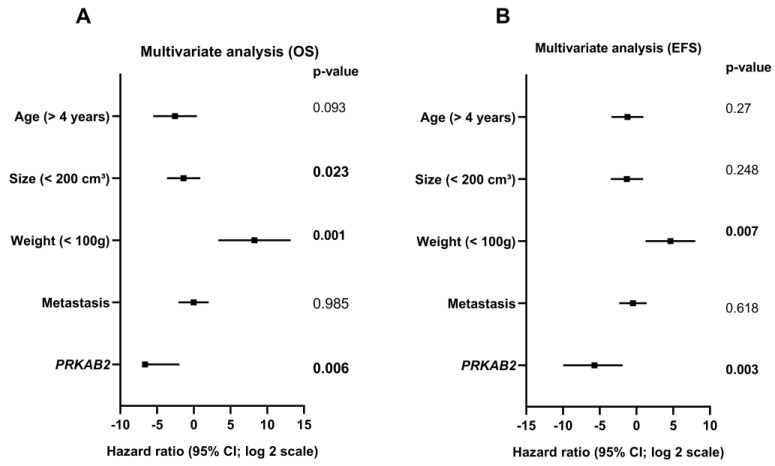
Multivariate analysis using the Cox regression model according to overall (OS) and event-free survival (EFS). (**A**) Overall survival: patient’s age > 4 years (*p* = 0.093), tumor size < 200 cm^3^ (*p* = 0.023), tumor weight < 100 g (0.001), presence of metastasis (*p* = 0.985), and lower *PRKAB2* gene expression (*p* = 0.006). (**B**) Event-free survival: patient’s age > 4 years (*p* = 0.27), tumor size < 200 cm^3^ (*p* = 0.248), tumor weight < 100 g (*p* = 0.007), presence of metastasis (*p* = 0.618), and lower *PRKAB2* gene expression (*p* = 0.003).

**Figure 4 cancers-16-01094-f004:**
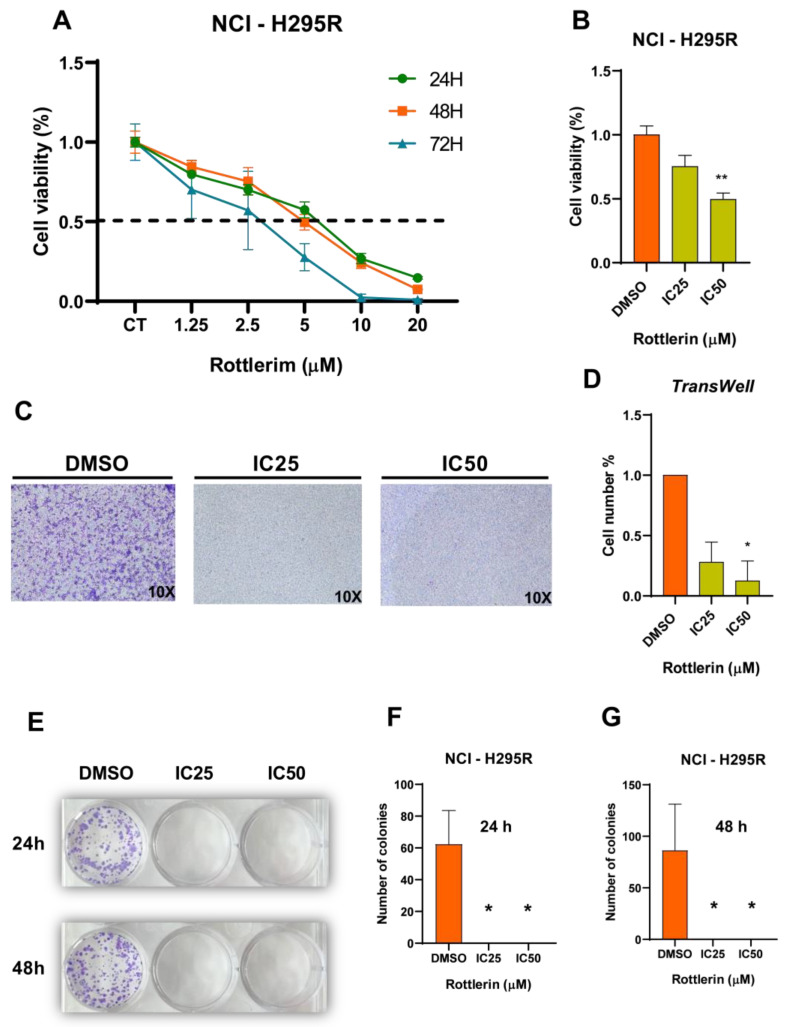
(**A**) Cell viability curve for NCI-H295R cells after treatment with Rottlerin for 24, 48, or 72 h to calculate IC25 and IC50. (**B**) Rottlerin decreased cell viability at IC50 (** *p* = 0.0034). (**C**) Transwell test with DMSO or Rottlerin at IC_50_ or IC_25_ (**D**). Rottlerin at IC_50_ decreased cell migration (* *p* = 0.025). (**E**–**G**) Clonogenic assay using DMSO or Rottlerin at IC_50_ or IC_25_ showed that clonogenic capacity was reduced after treatment with Rottlerin for 24 h and 48 h at IC_50_ (* *p* = 0.0102) or IC_25_ (* *p* = 0.0102) (ANOVA test).

**Figure 5 cancers-16-01094-f005:**
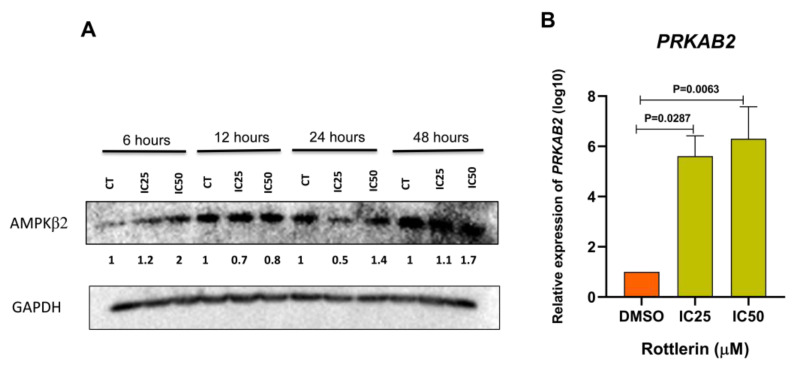
(**A**) Rottlerin-induced AMPKβ2 protein overexpression in NCI-H95R cells after treatment for 48 h. The uncropped blots are shown in Appendix A. (**B**) Rottlerin-induced overexpression of the *PRKAB2* gene in NCI-H95R cells after treatment for 48 h at IC_50_ (*p* = 0.0063) or IC_25_ (*p* = 0.0287).

**Figure 6 cancers-16-01094-f006:**
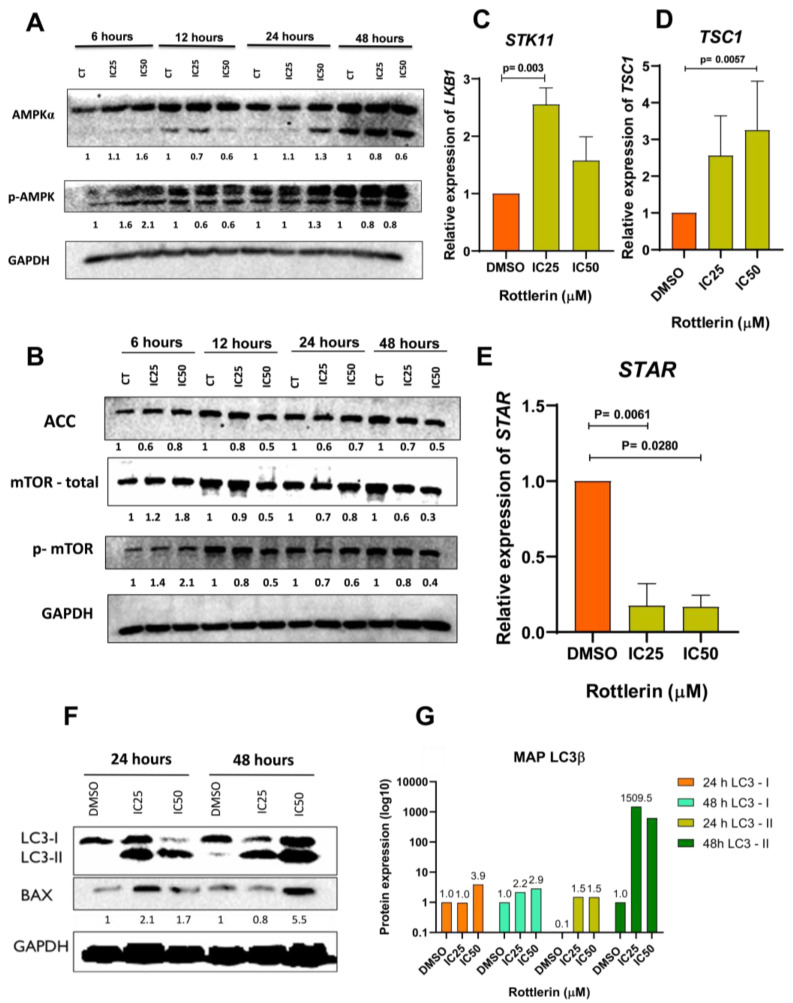
(**A**) Rottlerin increased the levels of total and Thr-172-phosphorylated AMPK in NCI-H295R cells. The percentage was greater after treatment with Rottlerin at IC_50_ for 6 h. (**B**) Treatment with Rottlerin at IC_50_ or IC_25_ for 12, 24, or 48 h reduced mTOR and p-mTOR activity in NCI-H295R cells, but mTOR activity was higher after treatment with Rottlerin at IC_50_ or IC_25_ 6 h. (**C**) Rottlerin reduced *STK11* gene expression in NCI-H295R cells (*p* = 0.0033). (**D**) Rottlerin at IC_50_ increased *TSC1* gene mRNA (*p* = 0.0141) in NCI-H295R cells. (**E**) Rottlerin at IC_50_ or IC_525_ decreased STAR steroid factor gene expression (*p* = 0.0280 and 0.0061, respectively) in NCI-H295R cells. (**F**) Treatment with Rottlerin for 24 or 48 h induced the protein expression of autophagy (MAP LC3beta) and apoptosis (Bax) markers. (**G**) Protein quantification of the LC3beta autophagy marker. The uncropped blots are shown in Appendix A.

**Figure 7 cancers-16-01094-f007:**
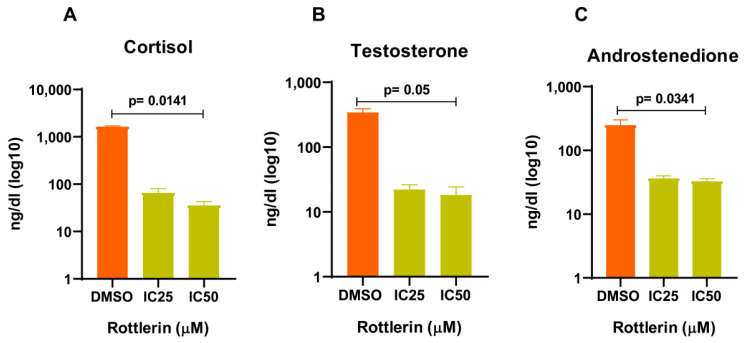
Hormone dosage after treatment with Rottlerin at IC_50_ or IC_25_. (**A**) The cortisol level was significantly reduced, and the reduction was more significant after treatment with Rottlerin at IC_50_. (**B**) The testosterone level was reduced, without statistical significance between the Rottlerin concentrations. (**C**) The androstenedione level was significantly reduced, and the reduction was more significant after treatment with Rottlerin at IC_50_. The Two-Way ANOVA test was used.

**Figure 8 cancers-16-01094-f008:**
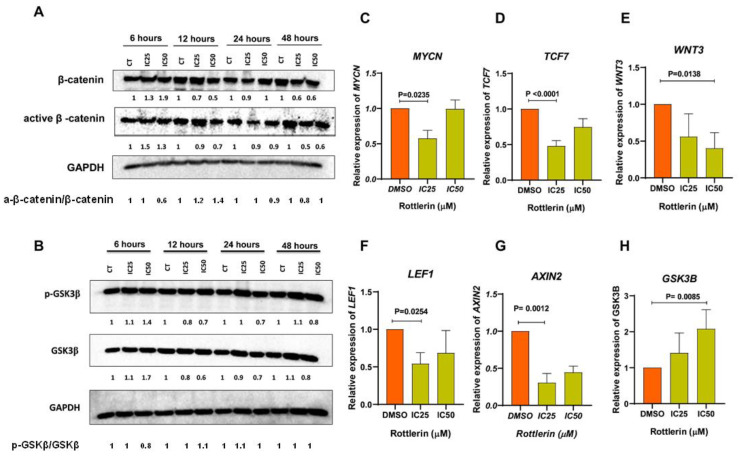
(**A**) Treatment with Rottlerin at IC_50_ or IC_25_ for 12, 24, or 48 h inhibited the β-catenin and β-catenin-activated protein expression levels in NCI-H295R cells. (**B**) NCI-H295R cells treated with Rottlerin for 6 h showed slightly increased p-GSK3β and total GSK3B, but treatment for 12, 24, or 48 h decreased the concentrations of these proteins. (**C**) Treatment with Rottlerin at IC_25_ decreased *MYC* gene expression in NCI-H295R cells (*p* = 0.0012). (**D**) Treatment with Rottlerin at IC_25_ decreased *TCF7* gene expression in NCI-H295R cells (*p* < 0.0001). (**E**) Treatment with Rottlerin at IC_25_ decreased *TCF7* gene expression in NCI-H295R cells (*p* = 0.0006). (**F**) Treatment with Rottlerin at IC_25_ decreased *LEF1* gene expression in NCI-H295R cells (*p* = 0.0235). (**G**) Treatment with Rottlerin at IC_25_ decreased *Axin2* gene expression in NCI-H295R cells (*p* < 0.0001). (**H**) Treatment with Rottlerin at IC_50_ for 48 h increased *GSK3B* gene expression in NCI-H295R cells (*p* = 0.0085). The uncropped blots are shown in Appendix A.

**Figure 9 cancers-16-01094-f009:**
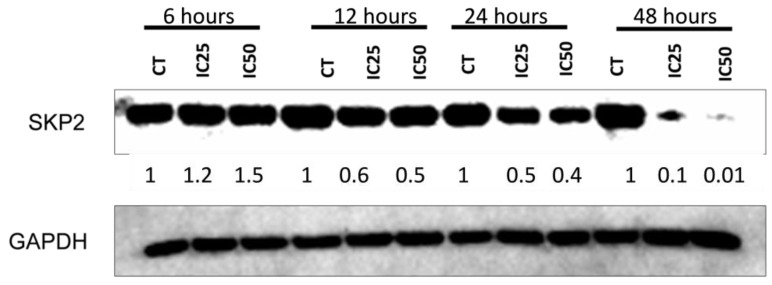
Treatment with Rottlerin for 12, 24, or 48 h reduced the SKP2 protein level in NCI-H295R cells. The uncropped blots are shown in Appendix A.

**Figure 10 cancers-16-01094-f010:**
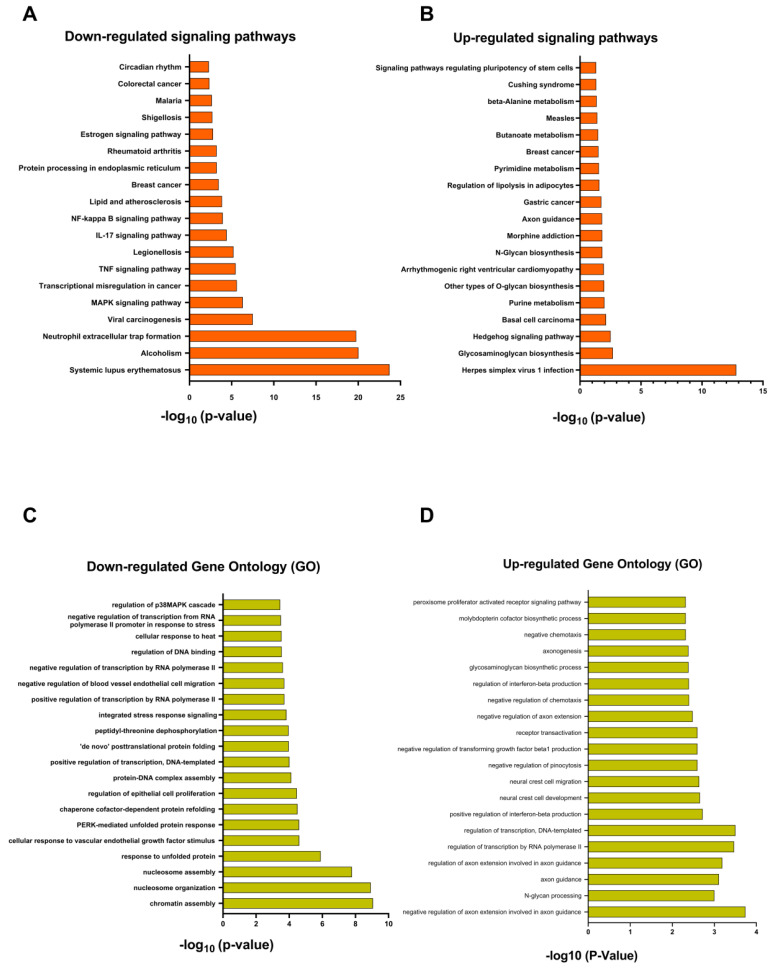
(**A**) In orange, positively regulated pathways, with *p* < 0.05. (**B**) In green, gene ontology, positively regulated pathways, with *p* < 0.05. (**C**) In orange, negatively regulated pathways, with *p* < 0.05. (**D**) In green, gene ontology, negatively regulated pathways, with *p* < 0.05. Pathway enrichment was analyzed by using the Enrichr platform (https://maayanlab.cloud/Enrichr/ (accessed on 23 February 2023)).

**Figure 11 cancers-16-01094-f011:**
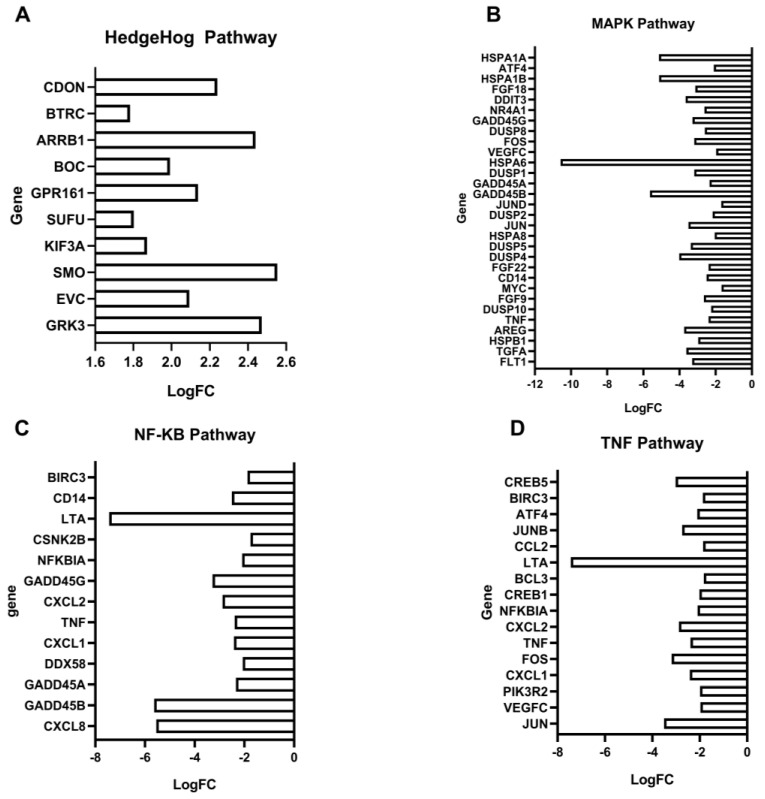
Pathway genes enriched during RNAseq analysis. (**A**) Hedgehog pathway gene expression level (*p* = 0.003) as LogFC. (**B**) MAPK pathway gene expression level (*p* < 0.001) as LogFC. (**C**) NF-KB pathway gene expression level (*p* = 0.002) as LogFC. (**D**) TNF gene expression level (*p* < 0.001) as LogFC.

**Table 1 cancers-16-01094-t001:** Analysis of five-year overall (5y-OS) and five-year event-free (5y-EFS) survival rates according to the clinical characteristics.

Variables	*n*	5y-EFS (%)	*p*-Value	5y-OS (%)	*p*-Value
Age	63		0.032		0.006
<4 years	45	82.2%		91.1%	
>4 years	18	55.6%		61.1%	
Gender	63		0.228		0.503
Male	13	61.5%		76.9%	
Female	50	78.6%		84%	
Size	63		0.001		<0.001
<200 cm³	50	84.0%		92%	
>200 cm³	13	38.5%		46.2%	
Weight *	62		0.001		<0.001
<100 g	40	87.5%		97.5%	
>100 g	22	50.0%		54.5%	
*p. R337h*	63		0.267		0.150
Present	54	72.2%		79.6%	
Absent	9	88.9%		100%	
Modified Sandrini	63				
I	38	89.5%	<0.001	94.4%	<0.001
II	14	64.3		71.4%	
III	3	100%		100%	
IV	8	12.5%		25%	
Metastasis	63		<0.001		<0.001
Present	7	14.6%		28.6%	
Absent	56	82.1%		89.3%	

* A patient had no record of tumor weight. EFS—event-free survival; OS—overall survival.

## Data Availability

The data presented in this study are available in this article and Appendix A.

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
