# Peer review of "Low *PRKAB2* Expression Is Associated with Poor Outcomes in Pediatric Adrenocortical Tumors, and Treatment with Rottlerin Increases the *PRKAB2* Level and Inhibits Tumorigenic Aspects in the NCI-H295R Adrenocortical Cancer Cell Line"

_cancers, 2024, doi:10.3390/cancers16061094_

Round 1
Reviewer 1 Report (Previous Reviewer 2)
Comments and Suggestions for Authors
The manuscript received significant ameliorations and I maintained my previous general evaluations. It can be accepted.
Author Response
Thank you very much for the review and comments.
Reviewer 2 Report (New Reviewer)
Comments and Suggestions for Authors
In this paper, the authors determine that the level of PRKAB2 expression correlates with the overall and free event survival of patients with pediatric adrenocortical tumors. Moreover, they show a decrease of H295 cell survival, migration and clonogenicity in vitro after treatment with Rottlerin that modulates PRKAB2 expression. This study is interesting but I have the following comments:
- In the introduction, line 67 to 75, it is not clear if the data have been previously published knowing that the GSE182022 is not available yet or if they are the ones from this paper.
- The authors did not explain how did they define the threshold of PRKAB2 expression value to generate the two groups with high and low PRKAB2 expression for overall survival and event free survival both in their and COG cohort. Did they use the same value on both cohort?
- For the clonogenic experiment, did you see cells that adhere to the plates while reseeding them after their treatment with Rottlerin? Indeed, it is surprising to see no colonies at all. Did they try the clonogenic experiment under Rottlerin treatment but without reseeding the cells through the experiment?
- AMPKb2 expression increases overtime in the control also in western blot, it is therefore difficult to draw a conclusion on the effect of Rottlerin on AMPKb2 expression. Moreover, the authors indicate the quantification of AMPKb2 expression compared to each control but not comparing to others time point. It could be interesting to express the ratio of AMPKb2 expression in treated condition to control overtime.
- Did the authors calculate the ratio phosphorylated AMPKb2 on total AMPKb2 for each condition? Similar analysis could be done for p-mTOR/mTOR, pGSK3/GSK3 and active b-catenin/b-catenin. This could facilitate the interpretation of the consequences of Rottlerin on AMPKb2, mTOR, active b-catenin/b-catenin.
- Figure 8: How long have been treated H295R cells with Rottlerin for LEF1 and AXIN2 expression? How do the authors explain that their expression decreases after Rottlerin treatment at IC25 and not at IC50?
- Line 525: The authors state that 6h Rottlerin treatment at IC25 and IC50 increases GSK3b and p-GSK3b and declined after but on the WB, it is difficult to determine if these differences and especially the increase at IC25 are significant. How many times this experiments have been reproduced and did the authors do statistical test on the western blot quantification for this and others experiments?
- Even though the authors put in perspectives the signaling results in the discussion, they should also add a little context on the reasons to look at the different signaling pathway in response to Rottlerin treatment (mTOR, Wnt/b-catenin signaling, SKP2) in the results section to facilitate the comprehension of the results.
Minor comments:
- Legend of the figure 1: The legend E does not match the graph of modified Sandrini explained in legend of F. The legend E should be removed or the authors should show the panel.
- Figure 8: The panel D and E and G and H overlap and should be separated.
- Are the western blot quantifications written below the images the direct band intensity or the ratio protein of interest to housekeeper?
Comments on the Quality of English Language
No specific comment on the English.
Author Response
Reviewer 2
Thank you for your relevant comments and suggestions. All changes are marked in blue in the revised manuscript.
In this paper, the authors determine that the level of PRKAB2 expression correlates with the overall and free event survival of patients with pediatric adrenocortical tumors. Moreover, they show a decrease of H295 cell survival, migration and clonogenicity in vitro after treatment with Rottlerin that modulates PRKAB2 expression. This study is interesting but I have the following comments:
- In the introduction, line 67 to 75, it is not clear if the data have been previously published knowing that the GSE182022 is not available yet or if they are the ones from this paper.
Response: Thanks for your comment. I would like to inform you that GSE182022 is not yet publicly available for consultation, as the article resulting from this study is still in the final stages of preparation. Once accepted, access will be made available. Data supporting study results may be made available upon reasonable request.
- The authors did not explain how did they define the threshold of PRKAB2 expression value to generate the two groups with high and low PRKAB2 expression for overall survival and event free survival both in their and COG cohort. Did they use the same value on both cohort?
Response: The cutoff point used for segmenting the gene expression of PRKAB2 was the median, as described in the Materials and Methods section during statistical analyses in both cohorts. Please see lines 266-267 and 291-292
- For the clonogenic experiment, did you see cells that adhere to the plates while reseeding them after their treatment with Rottlerin? Indeed, it is surprising to see no colonies at all. Did they try the clonogenic experiment under Rottlerin treatment but without reseeding the cells through the experiment?
Response: The initial phase of the clonogenic experiment involved seeding the cells followed by treatment after fixation, but it did not result in the formation of colonies. In a subsequent approach, we modified the protocol, treating cells with rottlerin for 24 and 48 hours, followed by seeding only viable cells. However, we did not observe colony formation even with this modification, leading us to proceed with this methodology throughout the experiment, as the outcome was consistent in both techniques.
- AMPKb2 expression increases overtime in the control also in western blot, it is therefore difficult to draw a conclusion on the effect of Rottlerin on AMPKb2 expression. Moreover, the authors indicate the quantification of AMPKb2 expression compared to each control but not comparing to others time point. It could be interesting to express the ratio of AMPKb2 expression in treated condition to control overtime.
Response: We appreciate your feedback. We performed four treatments at different time intervals (6, 12, 24, and 48 hours), each with its respective control. In all cases, the treatment was normalized to GAPDH and relativized to its specific time control. This approach was adopted due to the nature of the AMPK pathway, serving as a bioenergetic sensor in cells. Given the potential variation in its activity over the culture period (resulting from increased nutrient consumption in the culture medium), we deemed it more prudent to compare only between control and treated groups at the same time point.
- Did the authors calculate the ratio phosphorylated AMPKb2 on total AMPKb2 for each condition? Similar analysis could be done for p-mTOR/mTOR, pGSK3/GSK3 and active b-catenin/b-catenin. This could facilitate the interpretation of the consequences of Rottlerin on AMPKb2, mTOR, active b-catenin/b-catenin.
Response: Thank you for the observation. We added the results of the ratio of phosphorylated and total proteins to the respective figures as suggested.Please see Figure 6
- Figure 8: How long have been treated H295R cells with Rottlerin for LEF1 and AXIN2 expression? How do the authors explain that their expression decreases after Rottlerin treatment at IC25 and not at IC50?
Response: Thank you for considering the point raised. Our hypothesis is based on dosage disparity and its effects at the molecular level. However, we lack support in the literature to substantiate or explain the observed phenomenon. Therefore, additional studies are needed to improve understanding of the phenomenon.
- Line 525: The authors state that 6h Rottlerin treatment at IC25 and IC50 increases GSK3b and p-GSK3b and declined after but on the WB, it is difficult to determine if these differences and especially the increase at IC25 are significant. How many times this experiments have been reproduced and did the authors do statistical test on the western blot quantification for this and others experiments?
Response: We appreciate the observation. We did not perform statistical analysis for the protein quantifications, since the WB experiments were done in single replicates
- Even though the authors put in perspectives the signaling results in the discussion, they should also add a little context on the reasons to look at the different signaling pathway in response to Rottlerin treatment (mTOR, Wnt/b-catenin signaling, SKP2) in the results section to facilitate the comprehension of the results.
Response: In the discussion section, we address the status of each pathway in adrenocortical tumors. Please see lines 540-592
Minor comments:
- Legend of the figure 1: The legend E does not match the graph of modified Sandrini explained in legend of F. The legend E should be removed or the authors should show the panel.
- Figure 8: The panel D and E and G and H overlap and should be separated.
Response: They were corrected according to your suggestion
Reviewer 3 Report (New Reviewer)
Comments and Suggestions for Authors
The article is quite interesting, but in its current form, it requires numerous corrections and is not suitable for publication.
Justification:
The English requires intensive editing, with many unprofessional expressions not suitable for scientific texts. I suggest sending the article for professional language editing.
The authors state, "The present study analyzed PRKAB2 gene expression in 63 pediatric adrenocortical samples after microdissection." Are you sure it was microdissection? This method mainly applies to samples taken from histological preparations.
There is no information on how the material was collected and preserved.
It is necessary to expand the patient characterization data significantly. What were the values of measured hormones, and did they respond to the applied therapy, etc.?
There is no information about the qualitative tests used, such as those presented in Table 1.
One-way or two-way ANOVA was used for comparisons. Has it been checked whether the data meet the assumptions of the test? They should be verified. Why was no post-hoc test conducted? According to statistical assumptions, after obtaining statistically significant ANOVA results, a post-hoc test should be conducted to indicate differences between groups.
The sentence "The standard curve was constructed by considering an efficiency of 90–100%" is incorrect. The standard curve is constructed to determine efficiency.
Hormone concentrations were determined using RIA - a detailed description of the method is missing.
The authors state, "On the basis of LogFC, the differentially expressed genes (DEGs) were separated into high and low expression." It should rather be up and down-regulated genes.
What were the cutoff values for differentially expressed genes (fold change, adj. p val)?
Transcriptomic experiments should be presented from the general gene expression profile through volcano plots.
Raw transcriptomic data must be available in a publicly accessible repository, e.g., GEO. The publication must include the access number for this data.
In survival curves, I assume that survival for patients with low and high expression of PRKAB2 was shown - how were the groups divided?
In Figure 1A and B, the legend seems to indicate that up is unfavorable, while in Figure 1C - clinical outcome, higher concentrations are more favorable for patients. The same observation applies to Figure 2.
In Figure 3A, it seems that there should be no statistically significant values for size < 200 cm3 because the hazard ratio includes 0. Additionally, on this graph, the x-axis label should be Hazard ratio (95% CI; log2 scale).
There is no description of the numerical values in the Western blot images; moreover, decimal values should be separated by ".", not ",".
In Figure 11, besides fold values, the adj p value for each gene should also be provided
Comments on the Quality of English Language
The English requires intensive editing, with many unprofessional expressions not suitable for scientific texts. I suggest sending the article for professional language editing.
Author Response
Reviewer 3
Thank you for your relevant comments and suggestions. All changes are marked in blue in the revised manuscript.
Comments and Suggestions for Authors
The article is quite interesting, but in its current form, it requires numerous corrections and is not suitable for publication.
Justification:
The English requires intensive editing, with many unprofessional expressions not suitable for scientific texts. I suggest sending the article for professional language editing.
Response: Thank you for your observation. English was correted by a professional language editing.
The authors state, "The present study analyzed PRKAB2 gene expression in 63 pediatric adrenocortical samples after microdissection." Are you sure it was microdissection? This method mainly applies to samples taken from histological preparations.
Response: We appreciate your comment. Yes, we confirm that the employed technique was microdissection. We utilized this approach before molecular analysis (DNA, RNA, or protein extraction) to ensure the removal of potential necrotic areas from the post-resection tissue. A better description of this point was added in the revised form, please see lines 110-115
There is no information on how the material was collected and preserved.
Response: The tumor fragment was promptly frozen after surgical resection and stored in liquid nitrogen until needed. In the event of transportation, the entire process is carried out using cryopreservation. A better description of this point was added in the revised form, please see lines 110-115
It is necessary to expand the patient characterization data significantly. What were the values of measured hormones, and did they respond to the applied therapy, etc.?
Response: A supplemental table was added with the clinical data of all patients. Please see supplemental table 1
There is no information about the qualitative tests used, such as those presented in Table 1.
One-way or two-way ANOVA was used for comparisons. Has it been checked whether the data meet the assumptions of the test? They should be verified. Why was no post-hoc test conducted? According to statistical assumptions, after obtaining statistically significant ANOVA results, a post-hoc test should be conducted to indicate differences between groups.
Response: Thank you for the raised point, our description really was incomplete. To analyzed the association between gene expression and clinical characteristics we used the Mann-Whitney test since they did not present normal distribution in the analysis using the Kolmogorov-Smirnov test. To functional studies we used ANOVA and post-hoc Bonferoni test. We added this information in the revised form. Please see line 262
The sentence "The standard curve was constructed by considering an efficiency of 90–100%" is incorrect. The standard curve is constructed to determine efficiency.
Response: Thank you for the observation. You are right. It will be corrected! Please see lines 174-177
Hormone concentrations were determined using RIA - a detailed description of the method is missing.
Response: According to your suggestion we described better the methodology, please see lines 210-214.
The authors state, "On the basis of LogFC, the differentially expressed genes (DEGs) were separated into high and low expression." It should rather be up and down-regulated genes.
Response: Thank you, it will be corrected! Please see line 250-253
What were the cutoff values for differentially expressed genes (fold change, adj. p val)?
Response: We appreciate the question. A LogFC of 1.5 and a p-value < 0.05 were considered. We added this information in the revised form. Please see lines 250-253
Transcriptomic experiments should be presented from the general gene expression profile through volcano plots.
Response: Thank you for your observation. However, we specifically focused on AMPK interaction pathways. Evaluating the secondary pathways at this stage of the study would be less efficient and perhaps confusing, for this reason we believe that for this result this would be the best presentation.
Raw transcriptomic data must be available in a publicly accessible repository, e.g., GEO. The publication must include the access number for this data.
Response: This is being arranged, awaiting the GEO number. If the manuscript is accepted, it will be released for public consultation
In survival curves, I assume that survival for patients with low and high expression of PRKAB2 was shown - how were the groups divided?
Response: The cutoff point used for segmenting the gene expression of PRKAB2 was the median, as described in the Materials and Methods section during statistical analyses in both cohorts. Please see lines 266-267 and 291-292
In Figure 1A and B, the legend seems to indicate that up is unfavorable, while in Figure 1C - clinical outcome, higher concentrations are more favorable for patients. The same observation applies to Figure 2.
Response:Thank you for the observation. There was a nomenclature error, and it will be corrected. Please see Figure 1 and 2
In Figure 3A, it seems that there should be no statistically significant values for size < 200 cm3 because the hazard ratio includes 0. Additionally, on this graph, the x-axis label should be Hazard ratio (95% CI; log2 scale).
Response: Thank you for the observation. We have corrected it to log2.
There is no description of the numerical values in the Western blot images; moreover, decimal values should be separated by ".", not ",".
Response: Thank you for the observation. The highlighted point has been corrected.
In Figure 11, besides fold values, the adj p value for each gene should also be provided
Response: We appreciate the raised point. Our transcriptomic data were conducted in single replicates, and thus, we do not have adjusted p-values. We considered a p-value < 0.05 as significant.
Comments on the Quality of English Language
The English requires intensive editing, with many unprofessional expressions not suitable for scientific texts. I suggest sending the article for professional language editing.
Response: We appreciate your review. English was corrected by a professional language editing.
Reviewer 4 Report (New Reviewer)
Comments and Suggestions for Authors
The authors focus here on assessing whether there is a correlation between prkab2 gene overexpression and the clinical features of paediatric ACTs, and whether Rottlerin, a drug capable of acting on the AMP-cyclic pathway, may indeed be useful in this area. This natural drug-principle has already been evaluated but mainly for its ability to modulate Ca2+ channels and other types of channels at the cardiac level and for its ability to induce autophagy in particular cell subtypes (melanocytes). Seeing whether it might have other applications is interesting, and finding a use for it in a rare disease such as paediatric ACTs is very useful. However, this work has several critical issues that make it not suitable for publication in this form in CANCER.
Main issues:
Unclear purpose: the authors tend to describe what was done in the work but not the real purpose. The purpose itself needs to be rewritten
methods: methods are not described accurately. It is not possible to repeat experiments, reading them. It is not necessary to put all the information in the main methods, but if necessary, put them in a separate file as supplementary. For more detailed information see the comments directly in the attached revised manuscript. The supplementary table S1 should be mentioned in M&Met section, in RNA extraction section
English: English should be revised by native speakers with knowledge of scientific English
experimental set-up: if not previously published, the processing with conpusin should be made explicit and all concentrations and biological endpoints used for the calculation should be described in detail (again, if it is needed as a supplementary).
Special consideration should be done for WB: the raw images demonstrate that GHPDH used for normalization, for most of cases is at plateau. This means that it is not useful to really normalize expression of target genes and obtained result could be misleading, WB should be repeated with ad adequate normalization (even a total amount quantitation od proteins on sds-Page should/could be considered)
Comment on use of Rottlerin: I really appreciate the use of conpusin for determining IC50, however I do like prefer in this case to have a comment on Rottlerin dosed that can be used in humans. If IC50 is not safe in vivo, experiment using this dose in vitro on cell line, even id demonstrating a direct molecular effect could decrease rottlerin application for its unsafety
Comment on discussion section: some parts belong to introduction rather then discussion. Putting them in intro would simplify and shorten this long discussion. A critical approach to wb and RNA seq results on cell line should be envisaged. Are the same genes up/down regulated, if yes/no why? How are they linked to PRKAB2 in the the cAMP pathway?

Comments on the Quality of English Language
test should be revised
Author Response
Reviewer 4
Thank you for your relevant comments and suggestions. All changes are marked in blue in the revised manuscript.
The authors focus here on assessing whether there is a correlation between prkab2 gene overexpression and the clinical features of paediatric ACTs, and whether Rottlerin, a drug capable of acting on the AMP-cyclic pathway, may indeed be useful in this area. This natural drug-principle has already been evaluated but mainly for its ability to modulate Ca2+ channels and other types of channels at the cardiac level and for its ability to induce autophagy in particular cell subtypes (melanocytes). Seeing whether it might have other applications is interesting, and finding a use for it in a rare disease such as paediatric ACTs is very useful. However, this work has several critical issues that make it not suitable for publication in this form in CANCER.
Main issues:
Unclear purpose: the authors tend to describe what was done in the work but not the real purpose. The purpose itself needs to be rewritten
Response: Thank you for the comment, you are right. We rewrote the purpose according to your suggestion. Please see lines 95-102
methods: methods are not described accurately. It is not possible to repeat experiments, reading them. It is not necessary to put all the information in the main methods, but if necessary, put them in a separate file as supplementary. For more detailed information see the comments directly in the attached revised manuscript. The supplementary table S1 should be mentioned in M&Met section, in RNA extraction section
Response: A Supplementary file with more accurate description of the methodology was added to the revised form. The table is cited in this section, please see lines 168-169
English: English should be revised by native speakers with knowledge of scientific English
Response: English was correted by a professional language editing.
experimental set-up: if not previously published, the processing with conpusin should be made explicit and all concentrations and biological endpoints used for the calculation should be described in detail (again, if it is needed as a supplementary).
Response: Thank you for raising the issue. Compusyn is widely recognized and utilized software, as evidenced by searches in scientific articles. Below are some articles that also employed Compusyn for determining the IC50: 10.3390/molecules28134947; 10.1016/j.phymed.2021.153484
Special consideration should be done for WB: the raw images demonstrate that GHPDH used for normalization, for most of cases is at plateau. This means that it is not useful to really normalize expression of target genes and obtained result could be misleading, WB should be repeated with ad adequate normalization (even a total amount quantitation od proteins on sds-Page should/could be considered)
Response:We appreciate the observation. All proteins were quantified using the SDS-PAGE method, as described in the Results section. The identical values observed for GAPDH indicate an equivalent amount of protein per gel lane, making the data more comparable. We do not interpret the equality of bands as a plateau but rather as an indicator of a similar quantity of proteins.
Comment on use of Rottlerin: I really appreciate the use of conpusin for determining IC50, however I do like prefer in this case to have a comment on Rottlerin dosed that can be used in humans. If IC50 is not safe in vivo, experiment using this dose in vitro on cell line, even id demonstrating a direct molecular effect could decrease rottlerin application for its unsafety
Response: We appreciate your observation. So far, we have not identified clinical studies in humans with rottlerin. We are focusing our efforts on preclinical studies, where we aim to investigate the in vivo effect of rottlerin in an adrenocortical tumor xenograft model. Currently, there are studies with rottlerin in other tumors, using xenograft models, with a doses of 20mg/Kg-120mg/Kg (10.1038/s41564-021-00968-y and PMC3904376).
Comment on discussion section: some parts belong to introduction rather then discussion. Putting them in intro would simplify and shorten this long discussion. A critical approach to wb and RNA seq results on cell line should be envisaged. Are the same genes up/down regulated, if yes/no why? How are they linked to PRKAB2 in the the cAMP pathway?
Response: Thank you your considerations. Despite the slightly longer discussion, we believe that a more detailed description of the genes and pathways involved in this part of the text would be more didactic and facilitate the understanding and discussion of the results. According your suggestion we discussed better a critical aprouch of the RNAseq results and also the link between PRKAB2 and AMPK. Please see the lines 540-592.
Round 2
Reviewer 2 Report (New Reviewer)
Comments and Suggestions for Authors
I thank the authors for their answers. Here are my remaining comments:
- - About the clonogenic experiment, in your subsequent approach, while reseeding only viable cells, did you maintain cells under Rottlerin treatment? If yes, which concentration did you use and for how long did you wait for colonies formation? I think it would be good to perform the same experiment with lower Rottlerin concentration to determine the ranges of concentration efficient to prevent the clonogenic formation.
- - It will be interesting to redo the 6h Rottlerin treatment in order to be able to do statistical analysis on the treatment effect on GSK3b phosphorylation.
- - I think it will be useful to add information on the different signaling pathways analyzed in the results section to improve the readers understanding.
Author Response
The two initial suggestions, although interesting, would take a long time of weeks or months to carry out with the acquisition of new reagents, standardization and repetition of experiments and we believe that they would not change the results presented in a significant way, they would only add new information. The third suggestion had already been made by another reviewer and added to the latest version of the article.
- About the clonogenic experiment, in your subsequent approach, while reseeding only viable cells, did you maintain cells under Rottlerin treatment? If yes, which concentration did you use and for how long did you wait for colonies formation? I think it would be good to perform the same experiment with lower Rottlerin concentration to determine the ranges of concentration efficient to prevent the clonogenic formation.
Response: The cells were treated, and following the treatment, only viable cells were seeded and allowed to form colonies for 4 weeks. The cells were treated only once. We utilized the IC50 and IC25 values obtained after 48 hours in a 24-hour clonogenic experiment in an attempt to observe colony formation, which did not occur. Your suggestion to carry out the experiments with lower concentrations of Rotterin is interesting, however due to the time given, no additional experiments could be carried out.
- It will be interesting to redo the 6h Rottlerin treatment in order to be able to do statistical analysis on the treatment effect on GSK3b phosphorylation.
Response: We appreciate your suggestion. However, unfortunately, we do not have enough time available.
- I think it will be useful to add information on the different signaling pathways analyzed in the results section to improve the readers understanding.
Response: We appreciate your suggestion! However, the same suggestion has been made by another reviewer, and this point was added in the latest version of the manuscript.
Reviewer 3 Report (New Reviewer)
Comments and Suggestions for Authors
In its present form, the manuscript is much better, but the English and the quality of the figures still need to be improved. The size of the labels is too small and not very legible in many figures.
Comments on the Quality of English Language
In its present form, the manuscript is much better, but the English and the quality of the figures still need to be improved. The size of the labels is too small and not very legible in many figures.
This manuscript is a resubmission of an earlier submission. The following is a list of the peer review reports and author responses from that submission.
Round 1
Reviewer 1 Report
Comments and Suggestions for Authors
The Ms entitled "PRKAB2 as a prognostic biomarker and Rottlerin as a potential chemotherapeutic agent for pediatric adrenocortical tumor" by dr. Scrideli et coll. deals with the AdrenoCortical Carcinoma (ACC) (or AdrenCortical Tumors, ACT), a very rare disease with a dismal prognosis, with an impact on both pediatric and adult patients.
Although a deeper knowledge on the pathophysiology of this is needed as well as new therapeutical approaches, I do not judge the present Ms suitable for publication.
The Ms is incomprehensible, confused and not readable, with scientific and methodological important limits. It is difficult to understand the link of the two main topics of the Ms that are: PRKAB2 as a biomarker and the rottlerin as a potential “chemotherapeutic agent”.
Rottlerin up to nowdays is a compound for research. The rottlerin is a polyphenol natural product, that displays a complex spectrum of pharmacological properties. It has been used as a protein kinase C-δ (PKC-δ) inhibitor, although there has been some controversy in the literature over this claim (doi: 10.1016/j.tips.2007.07.003). The search on www.clinicaltrials.gov using rottlerin as a key-word leads 0 trials, so sentences related to the role of rottlerin as promising chemotherapeutic for treating ACTs written in the Title, Abstract and Conclusions are inappropriate.
The major criticism however is that methods and results shown did not allow to support the original hypothesis.
In detail:
PRKAB2 as a biomarker: although these results are interesting, the role of PRKAB2 as a biomarker needs to be validated in a prospective study
Although I do understand that NCI-H295R cells are the only commercially available ACC cell line, recently other experimental cell models become available publicly (doi: 10.3390/cancers15112873; doi: 10.1007/s12020-022-03112-w). Experiments done only with the NCI-H295R cell line cannot be acceptable, as results could be an epiphenomenon.
The IC-25 and IC-50 have been calculated in the concentration-response experiments and they should be in the Results section, not in the Methods. Then, Authors present results obtained with the IC-25 and IC-50 of rottlerin, that often present some discrepancies in relation to the concentration-dependent effect, however Authors never discussed these points.
The Results section reports unacceptable Western blots, both for the image resolution and for the technical aspects of the experiments themselves:
- Even in the supplementary data, only the strips with the bands of interest are shown, without showing the entire blot with an appropriate MW ladder, that is required by the Journal policy and it is respectful to reviewer and readers.
- Normalization with GAPDH has to be done for each experiment, while the GAPDH blots presented are used as normalizers of Western blots of different protein of interest (Fig.5, Fig.6A, Fig.8A; or Fig.8B, 6B,9B). If Authors stripped the membranes, this needed to be clarified. Further, I would like to underline that the Figures are incorrectly numbered and there is often non-English writing within the Figures. Finally, in none of the Figures both in the Ms and in the Supplementary data it is possible to identify the MW of the bands observed.
GAPDH in fig.6F is overexposed: how can be used it for normalization?
Paragraph 3.4 and Fig. 4: Fig.4A, 4F and 4G are redundant. I’m wondering whether Authors used an internal control to evaluate the migration capability of the NCI-H295R cells, as these cells usually did not display a very high capability of migration and invasion, due to their origin from a primitive ACC (doi: 10.1016/j.mce.2003.12.020; doi.org/10.3390/ijms24076829)
Paragraph 3.5: The increase of AMPKB2 expression of protein that reached its maximum after 6 hours rottlerin exposure is a very fast and unusual phenomenon that need to be characterized.
Comments on the Quality of English Language
It must be corrected
Reviewer 2 Report
Comments and Suggestions for Authors
Manuscript ID: cancers-2588926
Title: PRKAB2 as a prognostic biomarker and Rottlerin as a potential 2 chemotherapeutic agent for pediatric adrenocortical tumor.
In this study, the Authors evaluated the PRKAB2 as a prognostic biomarker for pediatric adrenocortical tumors and expanded the knowledge on the chemotherapeutic potentials of the well-known antitumoral natural compound Rottlerin in in vitro studies on the human NCI-H295R adrenocortical tumor cell line. The evaluation of various signaling pathways and expression of tumor-related genes/proteins corroborated their findings.
The manuscript is well-written and requires only the correction of a few minor points.
Minor points:
Figures 1 & 2: please translate the spanish word “censurado”.
Figure 2: correct “recorrence” with “recurrence”.
Figures 5 & 6: please translate the spanish word “horas”.
Figure 6 G: probably is better the use a different color or intensity for 48 hours.
Lines 303 & 449: “D4-androsteneidione” should be “D4-androstenedione”.
Line 451: “which can is evidenced by” should be “which can be evidenced by”.
Reviewer 3 Report
Comments and Suggestions for Authors
The authors of this study have investigated a cohort of pediatric ACC (I guess they are ACCs, but the authors refer to them as ACTs), and demonstrate reduced PRKAB2 gene expression which serves as a potential indicator of a poorer prognosis. Additionally, the administration of Rottlerin elicits several significant effects: it promotes PRKAB2 expression as well as the activation of total and phosphorylated AMPK, while inhibiting the mTOR, Wnt/β-catenin, and SKP2 pathways. Furthermore, Rottlerin stimulated autophagy and apoptosis and reduces cell viability, clonogenic potential, and migratory capacity in in vitro experiments. The work is original and of clinical interest. However, the m.s. could be polished in many aspects:
1. There is no information regarding the diagnosis of adrenal cortical carcinoma (ACC) for this patient cohort. Please include a statement regarding the histological and/or clinical criteria used to separate ACC from adenomas in this pediatric cohort. A germline TP53 p.R337H mutation itself is not sufficient in this aspect, as Li-Fraumeni patients also can develop benign adrenal cortical tumors. This is important to clarify, since 82% of the patients did not show distant metastases.
2. The H295R cell line is derived from an adult patient with ACC. Do we know the driver genes of this cell line, surely they must have been published (in for example PMID: 30870809)? If so, is this a TP53 mutated cell line? The reason for me asking is, would this cell line be the best choice for this study, and if so, please motivate this. Given the very high frequency of germline TP53 mutations in the clinical cohort, I assume the authors chose the cell line most closely resembling their own cases. Are there not other cell lines available as well (PMID: 35764904), be they commercial or not?
3. Figures need a lot of work as per my "minor issues" below. Many have annotations that are not even written in English. The most puzzling aspect is the occurrence of Fig 1-4 in the end of the m.s., while the text refer to figures 8-11? These types of mistakes makes the review of this paper harder.
4. The results seem robust, but I think the authors should reinforce that the clinical cohort used primarily is overrepresented in germline TP53 gene mutations. Although the study also analyzes data from the Children Oncology Group cohort, there is no specified information as to whether this ACC cohort has a more non-Brazilian distribution of germline mutations (i.e. no founder mutation effect). This discussion would improve the paper, and make the results more easily relatable for non-Brazilian researchers. Indeed, future, additional studies in pediatric ACCs without germline TP53 gene mutations would be interesting in terms of PRKAB2 expression.
Minor issues:
5. The nomenclature "pediatric adrenocortical tumor" is used and abbreviated as ACT according to the abstract, but then this abbreviation is used for "adrenocortical tumors" in the introduction. Please be consistent. Also, I believe the authors are referring to adrenocortical carcinoma, and not "tumor". Please revise accordingly. Statements such as "ACTs are rare and aggressive, and their incidence is estimated to lie between 0.7 and 2 41
cases/year per million inhabitants in adult patients" is not true if not specified that the authors refer to carcinomas.
6. The first sentence of the Introduction is rather redundant. "Adrenocortical tumors (ACTs) are a class of tumors that affect the adrenal cortex."
7. I lacked a few words in the Introduction section regarding what is known about AMP-activated protein kinase, PI3K/AKT/mTOR signalling and pediatric ACCs from previous literature. A few sentences regarding commonly dysregulated pathways (Wnt/TP53) would be good to introduce here as well.
8. Table 1 lists "Modified Sandrini staging", how does this correlate to ENSAT staging or AJCC/TNM?
9. Kaplan-Meier curves should be annotated with numbers at risk for each event.
10. Fig 1A, please do not use Portuguese (?) language in the figure. Fig 1C, use "alive/dead, not alive/death. Figs 1C-D-E are not aligned.
11. Fig 2, it must be specified in the legend that this is an independent cohort. Fig 2D (recurrence, not recorrence).
12. Fig 4A, are these results statistically significant? Please exand the results section in this aspect. Also, explain the asterisk in the legend. In Fig 1B and 1D, specifiy which comparisons that are significantly different by adding brackets to the staples as usually performed in these types of experiments. Fig 4C is really hard to appreciate, please enlarge this section.
13. Fig 5A, please use English only. Also, is it PRKAB2 protein (as in the legend), or AMPKbeta2 as in the figure itself? Be consistent.
14. Fig 6, again, English only. What is "ACC" in Fig 6B?
15. Fig 7B, testosterone is misspelled. Figures are poorly aligned.
16. Figure 4 (or is it 11??) - the last one anyway: Resolution is way too poor.
Comments on the Quality of English Language
See my comments in the report.